

# Dynamics of bacterial and archaeal communities during horse bedding and green waste composting

Vanessa Grenier[1,2], Emmanuel Gonzalez[3,4,5], Nicholas JB Brereton[6] and Frederic E. Pitre[1,2,7]

[1] Department of Biological Sciences, Université de Montréal, Montréal, Québec, Canada
[2] Institut de Recherche en Biologie Végétale, Montréal, Québec, Canada
[3] Department of Human Genetics, McGill University, Montréal, Québec, Canada
[4] Canadian Centre for Computational Genomics, McGill Genome Centre, McGill University, Montréal, Québec, Canada
[5] Gerald Bronfman Department of Oncology, McGill University, Montréal, Québec, Canada
[6] School of Biology and Environmental Science, University College Dublin, Dublin, Ireland
[7] Montreal Botanical Garden, Montréal, Québec, Canada

Corresponding author
Vanessa Grenier,
vanessa.grenier@umontreal.ca

## ABSTRACT

Organic waste decomposition can make up substantial amounts of municipal greenhouse emissions during decomposition. Composting has the potential to reduce these emissions as well as generate sustainable fertilizer. However, our understanding of how complex microbial communities change to drive the chemical and biological processes of composting is still limited. To investigate the microbiota associated with organic waste decomposition, initial composting feedstock (Litter), three composting windrows of 1.5 months (Young phase), 3 months (Middle phase) and 12 months (Aged phase) old, and 24-month-old mature Compost were sampled to assess physicochemical properties, plant cell wall composition and the microbial community using 16S rRNA gene amplification. A total of 2,612 Exact Sequence Variants (ESVs) included 517 annotated as putative species and 694 as genera which together captured 57.7% of the 3,133,873 sequences, with the most abundant species being *Thermobifida fusca*, *Thermomonospora chromogena* and *Thermobifida bifida*. Compost properties changed rapidly over time alongside the diversity of the compost community, which increased as composting progressed, and multivariate analysis indicated significant variation in community composition between each time-point. The abundance of bacteria in the feedstock is strongly correlated with the presence of organic matter and the abundance of plant cell wall components. Temperature and pH are the most strongly correlated parameters with bacterial abundance in the thermophilic and cooling phases/mature compost respectively. Differential abundance analysis revealed 810 ESVs annotated as species significantly varied in relative abundance between Litter and Young phase, 653 between the Young and Middle phases, 1182 between Middle and Aged phases and 663 between Aged phase and mature Compost. These changes indicated that structural carbohydrates and lignin degrading species were abundant at the beginning of the thermophilic phase, especially members of the Firmicute and Actinobacteria phyla. A high diversity of species capable of putative ammonification and denitrification were consistently found throughout the composting phases, whereas a limited number of nitrifying bacteria were identified and were significantly enriched within the later

mesophilic composting phases. High microbial community resolution also revealed unexpected species which could be beneficial for agricultural soils enriched with mature compost or for the deployment of environmental and plant biotechnologies. Understanding the dynamics of these microbial communities could lead to improved waste management strategies and the development of input-specific composting protocols to optimize carbon and nitrogen transformation and promote a diverse and functional microflora in mature compost.

# INTRODUCTION

The disposal of organic wastes in landfills has a negative impact on the environment due to the release of greenhouse gas and the pollution of soil, groundwater and surface water (*Lou & Nair, 2009*; *Taiwo, 2011*). Composting, which takes place in controlled environments allowing the maintenance of thermophilic conditions, can be a sustainable alternative to green waste disposal. Typical heat production results from the sequential action of various bacteria and fungi that degrade complex organic compounds, such as plant cell walls, into more accessible molecules (*Cragg et al., 2015*). The proliferation of these microorganisms is first determined by the nature of the composted material (*Vargas-García et al., 2010*; *Reyes-Torres et al., 2018*). Since microorganisms possess a range of specialized enzymes enabling the degradation of specific compounds, the availability and abundance of plant biomass constituents such as cellulose, hemicelluloses and lignin, as well as their availability and accessibility in the plant, will influence the recruitment of microorganisms during composting. Subsequently, environmental conditions such as temperature, ventilation and humidity will affect population dynamics and the rate and extent of organic matter decomposition (*Gajalakshmi & Abbasi, 2008*).

In large cities, green waste often makes up a significant portion of the municipal solid waste sent for composting. Most of this waste comes from the maintenance of public trees and green spaces, such as municipal parks and gardens, and includes tree and shrub cuttings as well as and grass clippings (*Reyes-Torres et al., 2018*). Lignocellulosic biomass is not only considered an organic waste, but also serves as a filler in compost. It provides a significant amount of dry matter and carbon to balance the high nitrogen and moisture content of food scraps and sewage sludge (*Haug, 1993*).

The complex and diverse nature of compost substrates, as well as the changing temperature and oxygen conditions within a defined environment, require the activity of equally complex and diverse communities of microorganisms to mineralize the organic matter (*Ryckeboer et al., 2003*; *Zhang et al., 2011*). Mesophilic actinobacteria such as *Kribbella* sp., *Actinoplanes* sp. and *Stackebrandtia* sp. and thermophilic actinobacteria such as *Mycobacterium* sp., *Thermobifida* sp., *Thermomonospora* sp. and *Thermobispor* a sp. are frequently found in aerobic compost, while the Firmicutes *Clostridium* sp., *Symbiobacterium*

sp., *Bacillus* sp. and *Geobacillus* sp. are commonly associated with anaerobic composts (*Antunes et al., 2016*; *Ryckeboer et al., 2003*; *Wang et al., 2016*). Thermophilic members of the Bacteroidetes and Chloroflexi, such as species within the genera *Rhodothermus* and *Sphaerobacter*, respectively, have also been reported in lignocellulose rich compost environments (*Antunes et al., 2016*). The concerted action of these multiple, sometimes synergistic, species is thought to enable the sequential release of carbon from biomass.

Further carbon conversion in composts may involve the formation of methane driven by methanogenic archaea, including thermophilic (*e.g.*, *Methanoculleus* sp. and *Methanosarcina* sp.) and mesophilic (*e.g.*, *Methanothermobacter* sp. and *Methanomicrobium* sp.) organisms which have previously been detected in compost (*Chen et al., 2014*; *Thummes, Kämpfer & Jäckel, 2007*; *Lee et al., 2010*).

Methanogens co-occur with methanotrophic or methane-oxidizing bacteria (MOB) in different ecosystems such as coastal/marine soils, rice fields, desert and forest soils (*Kumar et al., 2021*). Although the co-occurrence of methane-producing and methane-oxidizing communities has been described in composts made of manure and straw (*Chen et al., 2014*), most studies deal with the diversity and abundance of either methanogens (*Thummes, Kämpfer & Jäckel, 2007*) or methanotrophs (*Halet, Boon & Verstraete, 2006*), but rarely with the dynamics between the two groups.

Biological processes in compost rely on organic nitrogen supplied by organic materials such as plant residues, food waste, or manure (*Zhang et al., 2011*). Organic nitrogen mineralization, oxidation of ammonium and nitrite, and ammonia volatilization and denitrification, *i.e.,* the entire nitrogen cycle, occur at different stages of the composting process and is determined by the physicochemical conditions of the surrounding substrate (*Körner & Stegmann, 2002*). While ammonification is the predominant reaction in the early thermophilic stages, nitrification mostly occurs during maturation under the action of mesophilic ammonium-oxidizing bacteria (AOB) such as *Nitrosomonas* sp., *Nitrospira* sp., *Nitrosococcus* sp. and *Nitrosovibrio* sp. and nitrite-oxidizing bacteria (NOB) such as *Nitrospira* sp. and *Nitrobacter* sp. (*Körner & Stegmann, 2002*). Nitrogen losses through volatilization ($NH_3$) and denitrification (NO, $N_2O$ or $N_2$) are likely to occur through the action of microorganisms such as *Pseudomonas* sp., *Geobacillus* sp., *Bacillus* sp. and *Flavobacterium* sp. which can use nitrite and nitrate as a source of oxygen when anaerobic conditions prevail (*Verstraete & Focht, 1977*).

Considering the key role of the microbiota in the fundamental processes of lignocellulosic degradation and methane and nitrogen cycling during composting, this research aims to capture species-level changes in the microbial community at five time points, as well as the corresponding changes in physicochemical properties. Such detailed characterization is intended to contribute to the improvement of organic waste management through interventional approaches.

# MATERIALS & METHODS

## Study site, sampling, and physicochemical analyses

Three windrows (22 m × 5 m × 3 m) containing horse bedding (wood chips and horse manure) and green plant residues of varying maturity were sampled in summer 2018

(average site temperature = 27 °C). Residues in the youngest windrow were between 1 and 6 weeks old (Young), those in the second windrow were 3 months old (Middle), and those in the oldest windrow were 12 months old (Aged). All three windrows had been mixed with a tractor 2-3 times per month since they were put in place. Fresh horse bedding (Litter), the Young, Middle, and Aged piles, and a 24-month-old mature compost pile (Compost) were all sampled for analysis (Fig. 1) (*Grenier, 2021*). The three windrows were divided lengthwise into four sections and each section was sampled four times, taking two samples from each side of the windrow, while temperature was also measured at all 16 sampling points. Approximately 1 kg of material was collected at a depth of 60 cm at each sampling point, and the 4 samples from the same section were pooled to generate 4 composite samples per windrow ($n = 4$). As described in *Grenier (2021)*, four composite samples were collected at similar depths in the horse bedding pile and the mature compost pile. Samples were split and fractions were either frozen at −80 °C, refrigerated (4 °C), oven dried at 105 °C for 24 h, or air dried for two weeks. The oven- and air-dried samples were ground to a particle size of <2 mm before being stored in the dark until analysis. Organic matter was determined on the oven-dried samples by determining the weight loss after ignition at 600 °C. The oven-dried samples were used for analysis of the total carbon (C) and nitrogen (N) content by dry combustion at 950 °C using the varioMICROcube analyzer (Elementar, Langenselbold, Germany) and then extracted with water (1:10 (w/v)) for pH determination. Total mineral nitrogen ($NH_4^+$ and $NO_2^-$-$NO_3^-$) was measured on fresh samples extracted with 2.0M KCl (*Carter & Gregorich, 2006*) using methods for soil samples with the QuikChem® 8500 Series 2 FIA system (Lachat Instruments, Milwaukee, WI).

Cellulose, hemicellulose and lignin content were determined on air-dried samples with the ANKOM2000 Automated Fiber Analyzer (Ankom Tech., Macedon, NY, USA) according to the manufacturer's guidelines (http://www.ankom.com/procedures.aspx). Hemicellulose content was estimated as the difference between the neutral-detergent fiber (NDF) and the acid-detergent fiber (ADF), cellulose content was estimated as the difference between the ADF and the acid-detergent lignin (ADL) and lignin content was estimated as the difference between the ADL and the ash content.

## DNA extraction, PCR amplification, sequencing and processing

Total genomic DNA was extracted (250 mg of frozen samples) using Qiagen's DNeasy Power Soil® Pro kit and its quantity and quality were later determined spectrophotometrically using Thermo Fisher's NanoDrop 2000c. PCR reactions were performed using the forward primer P609D (5′- GGMTTAGATACCCBDGTA- 3′) and reverse primer P699R (5′-GGGTYKCGCTCGTTR-3′) targeting the V5-V6 region of the 16S ribosomal RNA (rRNA) gene and providing excellent coverage for bacterial and archaeal species (*Klindworth et al., 2013*). The amplification was performed under the following conditions: initial denaturation 94 °C for 2 min, denaturation 94 °C for 30 s, annealing 58 °C for 30 s, extension 72 °C for 30 s, final extension 72 °C for 7 min, 4 °C hold, over 35 cycles. The resulting amplicons (± 329 bp) were sequenced *via* Illumina MiSeq 2500 paired end 2 X

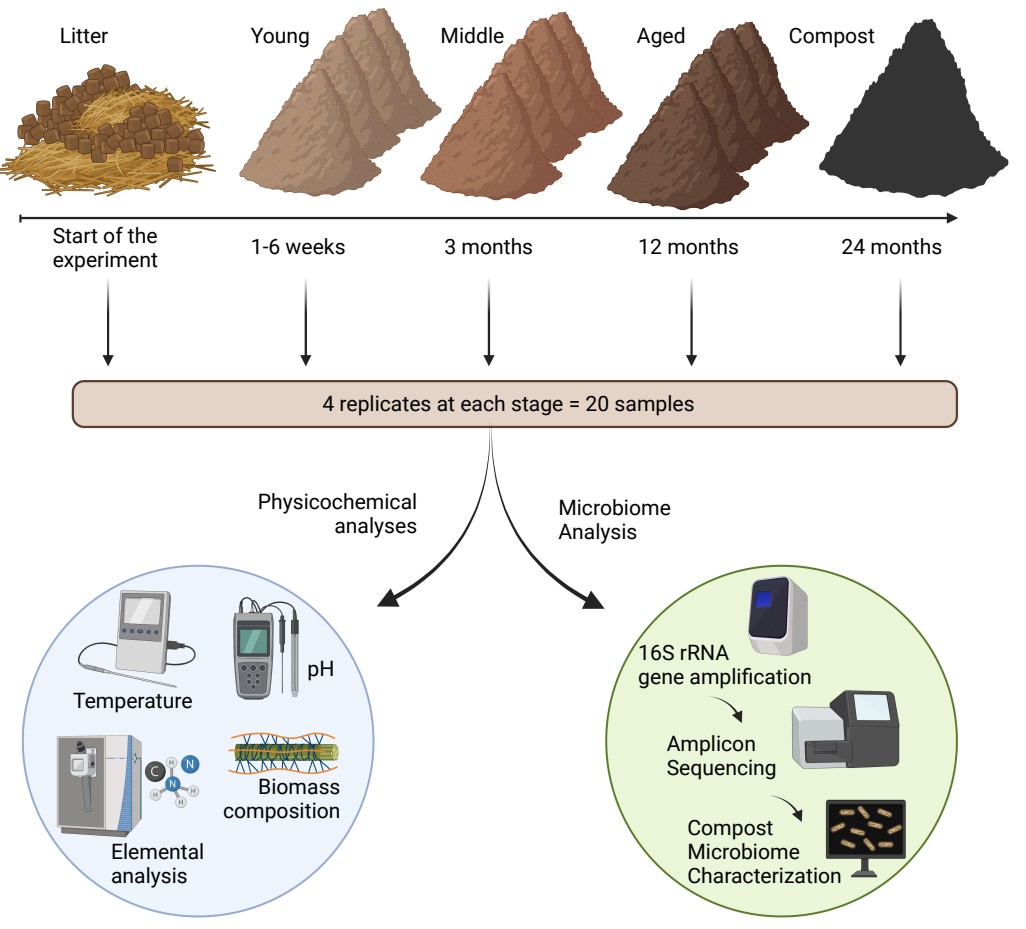

**Figure 1  Graphic representation of the sampling and analytical procedures.** This figure was made in © BioRender: biorender.com.

250 pb platform at the Genome Quebec Innovation Centre (Montreal, Canada). Reagent controls for quality assurance were below the detection limit.

The ANCHOR pipeline (*Gonzalez, Pitre & Brereton, 2019*) was used for the processing and annotation of sequence reads (https://github.com/gonzalezem/ANCHOR_v1.0). First, sequences were aligned and dereplicated using Mothur, then Exact Sequence Variants (ESVs) were selected based on a count threshold of 12 across all samples ($n = 20$). The annotation was performed with strict BLASTn criteria (99% identity and coverage) on 4 sequence repositories: NCBI-curated bacterial and Archaea RefSeq, NCBI nr/nt, SILVA and Ribosomal Database Project (RDP) (NCBI-curated bacterial and Archaea RefSeq is given a priority when at 100% identity and coverage). An *Ambiguous hit* annotation was retained and reported when multiple, equally good (highest identity/coverage), annotation was found. Amplicons with fewer than 12 counts across all samples were binned to high-count sequences in a second BLASTn, using a threshold of 98% identity/coverage. As databases are subjected to changes and updates, all annotations should be considered as assumptions and interpreted with caution.

## Statistical and differential abundance analysis

All statistical analysis for the physicochemical measurements were carried out in GraphPad Prism 8.4.3. The physicochemical data were tested for normal distribution using Shapiro–Wilk test (File S1) and the parameters that failed the test ($NO_2^-$-$NO_3^-$) underwent a square root transformation. One-way ANOVA followed by a Tukey's multiple comparisons post hoc test was used to compare physicochemical properties across successive composting phases. A Spearman correlation matrix presenting the interactions between physicochemical parameters (temperature, O.M., pH, total carbon, cellulose, hemicellulose and lignin content, total nitrogen $NH_4^+$ and $NO_2^-$-$NO_3^-$ content) and the 50 ESVs with the highest relative abundance over all five composting phases was produced to demonstrate the relationship between physicochemical parameters and microbial taxa dynamics. Alpha diversity was measured using Shannon index indices within Phyloseq package (*McMurdie & Holmes, 2013*). Alpha diversity was compared between the different groups of samples using a $t$-test. The Phyloseq package (*McMurdie & Holmes, 2013*) was used to perform a Redundancy Analysis (RDA) ordination based on Bray-Curtis ecological distances, while the veganCovEllipse function from Vegan package (*Oksanen et al., 2008*) in R (*R Core Team, 2021*) was used to draw the dispersion ellipses. Finally, the adonis function in the Vegan R Package was used to perform a PERMANOVA on the Bray distances matrices to evaluate the significant differences between the sampled compost piles. Differential abundance analysis on ESVs was performed using DESeq2 (*Love, Huber & Anders, 2014*; *Thorsen et al., 2016*), which was specifically conceived to offer good performance with uneven library sizes and sparsity (*Brereton, Pitre & Gonzalez, 2021*; *Gonzalez, Pitre & Brereton, 2019*; *Weiss et al., 2017*). A false discovery rate (FDR; Benjamini–Hochberg procedure) < 0.05 was applied (*Anders et al., 2013*; *Love, Huber & Anders, 2014*). Raw counts were log transformed across samples (rlog function, R Phyloseq package). Sparsity and low-count cut-offs were applied as ESV counts must be > 2 in 40% of the samples (*Dhariwal et al., 2017*; *Gonzalez, Pitre & Brereton, 2019*) while ESV count in a single sample is < 90% of the count in all samples.

The interpretation of the results focused on the identification of bacteria potentially involved in the transformation of carbon and nitrogen from lignocellulose. The association of bacterial species with potential functions is reported in the literature (File S2) but should be considered speculative as functions were not measured directly. To increase the confidence potential functional association, only ESVs identified at the species level, without ambiguous annotation and with an identity score of 100% were screened for roles in cellulose and lignin degradation, methane production and oxidation, ammonification, ammonium and nitrite oxidation and denitrification.

## RESULTS

### Physiochemical properties of the composting windrow

Physicochemical measurements were performed on the horse litter (Litter), three windrows of 1.5, 3 and 12 months of age (Young, Middle and Aged, respectively), and mature compost (Compost) (Fig. 2, raw data and complete statistical results in File S1). The

average temperature of the horse litter pile was 38.2 °C. Temperatures varied significantly ($p < 0.05$) between each successive compost phase, averaging 67.8 °C for Young, 62.1 °C for Middle, 46.1 °C for Aged and 37.0 °C for mature Compost. Organic matter content was 93.1% in the Litter and progressively reduced to 56.1% in Young, 42.9% in Middle, 31.7% in the Aged and 24.4% in the mature Compost. This decrease was significant between Litter and Young phase as well as between Young phase and Compost. The initial pH in the Litter was at 7.37, which increased significantly from pH 7.33 in the Young phase to pH 7.72 in Middle phase, and significantly increased to pH 8.42 in Aged phase and pH 8.23 in the mature compost.

Total carbon was 43.9% in Litter and decreased significantly to 26.9% in Young. Middle, Aged and Compost were similar, averaging 22.9%, 22.0% and 18.5% of carbon, respectively, with Compost being significantly lower than Young (Fig. 2, File S1). Nitrogen levels were at 1.37% in Litter before decreasing significantly to 1.09% in Young phase and remaining similar for Middle, Aged and Compost at 1.08%, 1.10% and 1.04%, respectively. The C:N ratio reduced significantly from 32.6 in the Litter to 22.7 in Young phase and then remained similar at 21.0 in Middle phase, 19.9 in Aged phase and 17.7 in the mature Compost.

Cellulose content dropped significantly from 32.20% (dry matter) in the Litter to 10.86% in the Young phase and then again to 6.19% in Middle phase, before reaching 5.50% in the Aged phase and 3.86% in the mature Compost (but was not significantly different compared to Middle phase) (Fig. 2). The hemicellulose content decreased significantly from Litter to Young, from 16.33% dry matter to 6.94%, and remained similar at 5.75% for Middle, 5.96% for Aged and 5.79% for the mature Compost. Lignin significantly decreased from 17.31% in Litter to 9.87% in Young phase. The lignin content of 7.79% in Middle, 7.99% in Aged and 6.49% in the mature Compost did not vary significantly from Young phase.

Ammonium ($NH_4^+$) content increased significantly from 16.45 mg kg$^{-1}$ in Litter to 31.07 mg kg$^{-1}$ in Young phase, and then dropped significantly from Young phase to 3.86 mg kg$^{-1}$ in Middle phase. The $NH_4^+$ content was 0.87 mg kg$^{-1}$ in Aged phase and 0.54 mg kg$^{-1}$ in Compost, significantly lower than Litter and Young phase. The $NO_2^-$-$NO_3^-$ content was of 10.11 mg kg$^{-1}$ in Litter, 13.70 mg kg$^{-1}$ in Young phase, 36.7 mg kg$^{-1}$ in Middle phase, 24.85 mg kg$^{-1}$ in Aged phase and 6.57 mg kg$^{-1}$ in Compost. The $NO_2^-$-$NO_3^-$ content increased between Litter and Middle phase and decreased significantly between the Middle phase and Compost phases.

**Composting microbial community**

A total of 3,133,873 amplicons were aligned with lengths (>0.1% counts) ranging 322-362 nt and 2,612 ESVs (Exact Sequenced Variants) were identified (File S2). Of these ESVs, 517 (19.8%) could be annotated as putative species, 694 (26.6%) could be annotated at the genera level, 486 (18.6%) at higher taxonomic levels and 915 (35.0%) were annotated as unknown (Fig. 3B). ESVs identified as putative species captured 32.3% of raw amplicon counts (Fig. 3C), had an average identity of 99.8%, including 186 ambiguous ESVs (similarity to multiple taxa). ESVs annotated at genus level captured 25.4% of raw counts, ESVs annotated at higher taxonomy level captured 25.3%, and

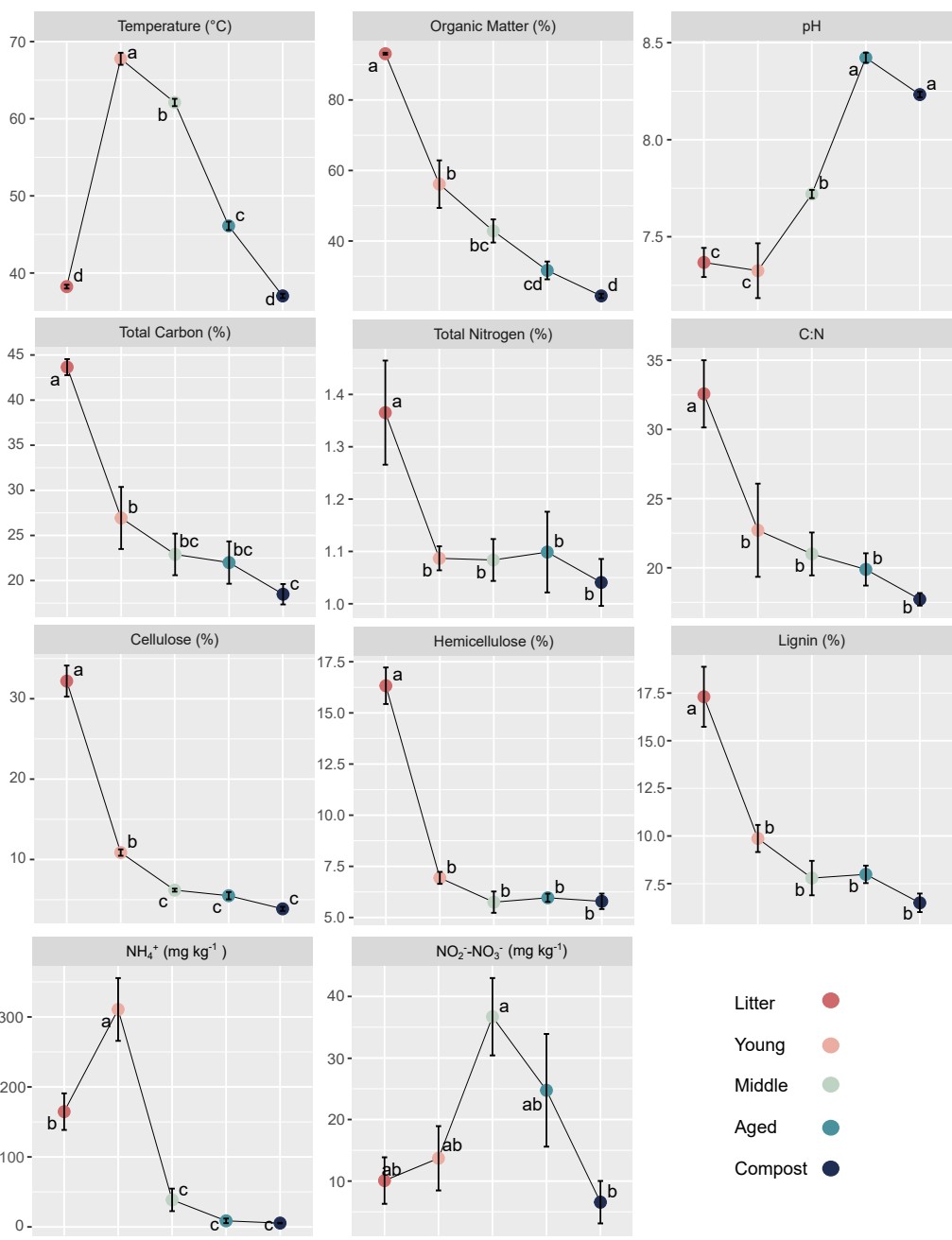

**Figure 2 Physicochemical properties at each stage of the composting process.** Temperature, organic matter, pH, plant cell wall composition and nitrogen fractions change through Litter, Young phase (1.5 months), Middle phase (3 months), Aged phase (12 months) and mature Compost (24 months). All values represent mean ($n = 4$) ± SD. The letters indicate significant differences between phases.

unknown ESVs captured 17.0%. ESVs identified across all phases belonged to 26 different phyla, of which Proteobacteria, Firmicutes, Actinobacteria, Bacteroidetes and Chloroflexi represent 56.0% of the total ESV diversity and shared 73.5% of the total raw counts,

while archaea represented 21 ESVs and 1.7% of total raw counts (Fig. 3A, stacked bar graph of relative abundance in File S2). Of the 517 ESVs identified to the species level, Thermobifida_fusca_2, Thermomonospora_chromogena_1 and Thermobifida_bifida_1 are the most abundant and account for 5.6% of the raw counts.

A Spearman correlation matrix plotted the physicochemical parameters measured in each phase against the relative abundance of the 50 most abundant species-identified ESVs (Fig. 4A, File S2). The 50 ESVs, sorted according to the phase in which they are most abundant (Fig. 4B), show a strong pattern suggesting that the abundant ESVs within a phase are largely correlated with the same physicochemical parameters. The highly abundant ESVs in Litter were strongly correlated with the amount of organic matter, including total carbon and nitrogen, plant cell wall components (cellulose, hemicellulose and lignin) and ammonium. ESVs were mostly not correlated with temperature and $NO_2^--NO_3^-$ content while they showed a strong negative correlation with pH. The correlation patterns were similar for ESVs abundant in the Young and Middle phases, which correspond to the thermophilic phases of the process. The relative abundance of ESVs was mainly correlated with temperature and $NH_4^+$ content for ESVs abundant in Young and temperature and $NO_2^--NO_3^-$ content for those abundant in Middle, while it was generally negatively correlated with OM, total carbon and nitrogen, and plant wall components. The correlation patterns were also similar for abundant ESVs in the Aged and Compost phases with a contrasting profile to the one observed in Litter. The relative abundance of ESVs was very negatively correlated with the amount of OM, plant cell wall components, total carbon and nitrogen, and $NH_4^+$ content. The eight relevant ESVs were, nevertheless, strongly correlated with pH, which was significantly higher in the Aged and Compost phases (Fig. 2).

Shannon's alpha-diversity index was significantly different between successive groups ($t$-test $p < 0.05$), except for Young phase vs. Middle phase (Fig. 3D) (Observed, Chao1, se.chao1, Shannon, Simpson, InvSimpson and Fisher indexes can be found in File S1). The lowest alpha diversity index was found in Litter and progressively increased at each phase with mature Compost having the highest index. Redundancy analysis (RDA) indicated that samples separated by time, with the first principal coordinate explaining 46.4% of the variance between the samples (Fig. 3E) and multivariate analysis identified significant variance between each phase (PERMANOVA, $p < 0.001$). A total of 810 ESVs were identified as significantly differently abundant (DESeq2, padj < 0.05) between the Litter and Young phase, 653 between the Young and Middle phases, 1182 between Middle and Aged phases and 663 between Aged phase and Compost (File S2).

## Differential abundance between Litter and Young phases

A total of 64 DA ESVs from Proteobacteria, 31 from Actinobacteria, 17 from Bacteroidetes and 15 from Firmicutes were observed as significantly reduced between Litter and Young phases (Fig. 3F), including 11 from the family Microbacteriaceae, seven from Bacillaceae, seven from Xanthomonadaceae and six from Alcaligenaceae. Reductions in the relative abundance of 64 DA ESVs annotated as putative species by a fold change of 5.8–8.0 were observed and included Sphingobacterium_jejuense_1,

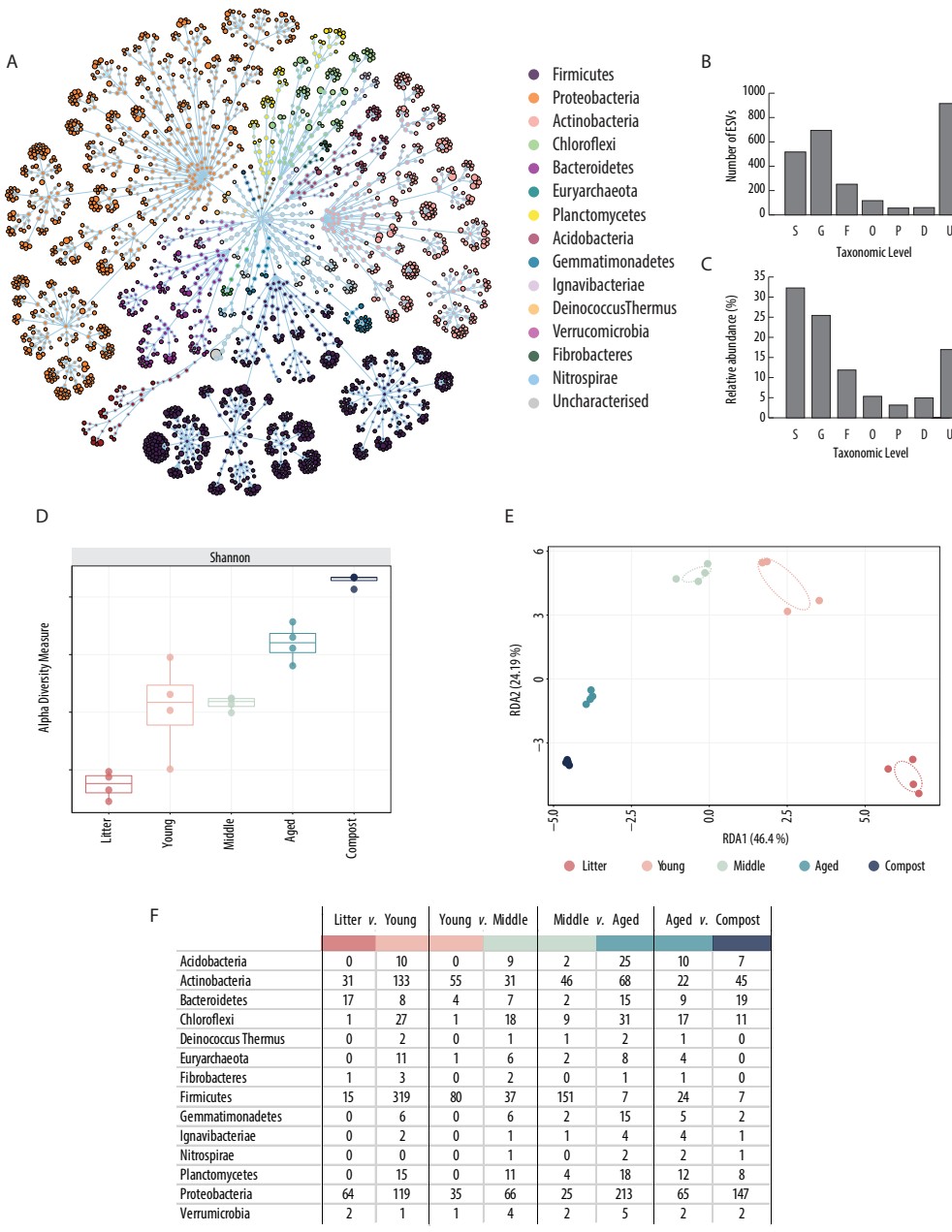

**Figure 3  Composting phases bacterial community overview.** (A) Flower diagram of the total microbial community throughout the five composting phases, (B) number of ESVs per taxonomic category (S = Species, G = Genus, F = Family, O = Order, C = Class, P = Phylum, D = Domain, U = Unknown), (C) percentage of ESVs per taxonomic category (S = Species, G = Genus, F = Family, O = Order, C = Class, P = Phylum, D = Domain, U = Unknown), (D) Shannon diversity indices of the bacterial communities in each composting phase, (E) redundancy analysis (RDA) of bacterial community in each phase, and (F) number of ESVs (classified by phylum) identified at the species level with 100% identity score and without ambiguous hit that are differentially abundant between Litter and Young, Young and Middle, Middle and Aged and Aged and Compost.



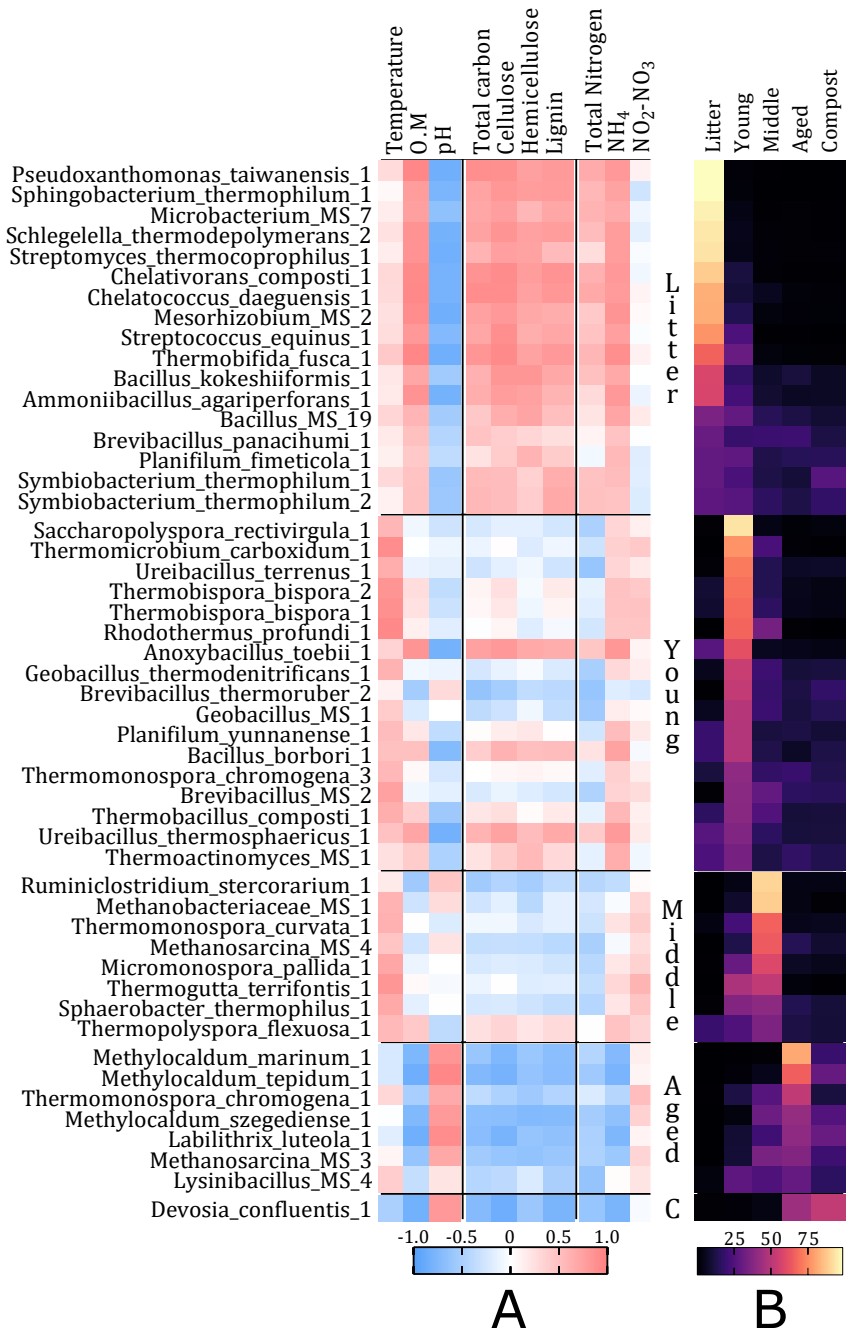

**Figure 4** **Spearman correlation matrix beween physicochemical parameters and the 50 ESVs with the highest relative abundance.** (A) Spearman correlation matrix showing the physicochemical parameters (temperature (T°C), O.M., pH, total carbon (%), cellulose (%), hemicellulose (%) and lignin (%) content, total nitrogen (%), $NH_4^+$ (mg kg$^{-1}$) and $NO_2^-$-$NO_3^-$ (mg kg$^{-1}$) content) and the 50 ESVs with the highest relative abundance over all five composting phases (ESVs are ranked according to their relative abundance in the different phases, so that the top ones are more abundant in the Litter phase, followed by the ones more abundant in the Young phase, *etc.*). (B) Matrix showing the relative abundance of the 50 ESVs in the different composting phases.

Delftia_litopenaei_1, Sandaracinus_amylolyticus_1, Nakamurella_panacisegetis_1 and Sphingobacterium_cibi_1 (File S2). A high number of DA ESVs increased in relative abundance from Litter to Young phases including 319 ESVs from Firmicutes, 133 from Actinobacteria, 119 from Proteobacteria and 27 from Chloroflexi. The largest amount of these significantly increased DA ESVs were from Firmicutes families (*e.g.*, Paenibacillaceae, Bacillaceae and Clostridiaceae), but the highest fold change increases included non-Firmicutes such as Thermomicrobium_carboxidum_1, Rhodothermus_profundi_1, Rhodothermus_marinus_1, Thermogutta_terrifontis_1, Thermus_thermophilus_1, Rhodothermus_marinus_2 and Thermoleophilum_album_1, which ranged between 11.0–13.6 fold higher in relative abundance in Young compared to Litter.

### Differential abundance between Young and Middle phases

DA ESVs observed as significantly lower in relative abundance in Middle compared to Young were dominated by 80 ESVs from Firmicutes, 55 from Actinobacteria and 35 from Proteobacteria, which included (*e.g.*, Bacillaceae, Microbacteriaceae, Limnochordaceae, Paenibacillaceae and Thermoactinomycetaceae) (Fig. 3F). Large reductions in relative abundance of 101 DA ESVs annotated as putative species included examples such as Jeotgalicoccus_psychrophilus_1, Corynebacterium_stationis_1, Corynebacterium _guangdongense_1, Corynebacterium_casei_1, Streptococcus_equinus_1, Kurthia _massiliensis_1, Weissella_confusa_1 which had reduced relative abundance by a fold change of 7.2−8.4 (File S2). The most highly represented phyla in DA ESVs which significantly increased in relative abundance from Young to Middle phases included 66 ESVs from Proteobacteria, 37 from Firmicutes, 31 from Actinobacteria, 18 from Chloroflexi and 11 from Planctomycetes. The DA ESVs significantly increasing in the Middle phase included examples such as Caenibacillus_caldisaponilyticus_1 and Methylocaldum_szegediense_1 as well as the Verrucomicrobia Limisphaera_ngatamarikiensis_1, which ranged from a fold change increase of 3.8–4.1. A total of six DA ESVs identified as putative archaea were also significantly increased in relative abundance from Young to Middle phase, including Methanothrix_thermoacetophila_1 and Methanoculleus_thermophilus_1 by a fold change of 5.7 and 1.8, respectively.

### Differential abundance between Middle and Aged phases

Changes between the Middle and Aged phases included substantial reductions of 151 DA ESVs from the phyla Firmicutes, (Paenibacillaceae and Bacillaceae), as well as 46 from Actinobacteria and 25 from Proteobacteria (Fig. 3F). From the 116 DA ESVs annotated as putative species, examples of substantial shifts included Limisphaera_ngatamarikiensis_1, Rhodothermus_profundi_1, Cohnella_laeviribosi_1, Thermomicrobium_carboxidum_1, Thermogutta_terrifontis_1 and Aneurinibacillus_thermoaerophilus_1, which reduced by 4.9−7.3- fold change (File S2). DA ESVs significantly increased in relative abundance from Middle to Aged phase included 213 ESVs from Proteobacteria, 68 from Actinobacteria, 31 from Chloroflexi, 25 from Acidobacteria, 18 from Planctomycetes, 15 from Bacteroidetes and 15 from Gemmatimonadetes. From the 49 DA EVSs annotated as putative species which significantly increased, large changes included two archaea, Nitrosotenuis_cloacae_1

and Methanosaeta_concilii_1, as well as Methylocaldum_marinum_1, Ignavibacterium_album_1, Syntrophobacter_sulfatireducens_1, Sulfurivermis_fontis_1 and Denitratisoma_oestradiolicum_1, which increase by 5.6–8.8-fold change.

## Differential abundance between Aged phase and mature Compost

DA ESVs which significantly reduced in relative abundance between Aged and mature Compost phases were dominated by 65 ESVs from Proteobacteria, 24 from Firmicutes, 22 from Actinobacteria and 17 from Chloroflexi, and were very widely distributed over 100 families, with the most frequent including eight ESVs from Bacillaceae, six from Caldilineaceae, five from Nitrosomonadaceae and five from Paenibacillaceae (Fig. 3F). From the 40 DA ESVs annotated as putative species, examples of substantial changes included Ignavibacterium_album_1, Methanothrix_thermoacetophila_1 (Archaea), Thermogutta_terrifontis_1, Leucobacter_chromiireducens_1, Rhodothermus_profundi_1 and Rhodothermus_marinus_2, which ranged from a reduction of between 4.4–6.5-fold change (File S2). DA ESVs which significantly increased in relative abundance between Aged and mature Compost phases were dominated by 147 ESVs from Proteobacteria, 45 from Actinobacteria, 19 from Bacteroidetes and 11 from Chloroflexi, including 51 ESVs from within the order Rhizobiales, such as Methylocystis_rosea_1, Mesorhizobium_tamadayense_1 and Rhizobium_helanshanense_1 which increased in relative abundance by between 3.7–4.4-fold change. From the 55 DA ESVs annotated as putative species, the largest changes included Methylosarcina_lacus_1, Flavobacterium_degerlachei_1, Lysobacter_yangpyeongensis_2 and Nitrospira_japonica_1, which ranged from a reduction of between 6.9–7.4-fold change.

## Microbes associated with carbon dynamics
### Lignocellulose degradation

Seventy-two differently abundant ESVs were annotated as species associated with putative cellulose degradation potential (cellulase or β-glucanase activity; Fig. 5, File S2). These included species within Firmicutes (34), Actinobacteria (24), Proteobacteria (6), Bacteroidetes (5), Chloroflexi (2) and Deinoccocus Thermus (1). In Young phase, 48 ESVs annotated as species associated with potential cellulose degradation significantly increased in relative abundance compared to Litter; these were dominated by ESVs from the order Bacillales (22), and included Ammoniphilus_resinae_2, Bacillus_borbori_1, Bacillus_coagulans_1, Brevibacillus_borstelensis_1, Brevibacillus_thermoruber_1 and _2, Cohnella_panacarvi_1, Geobacillus_stearothermophilus_2, Geobacillus_thermocatenulatus_1, Geobacillus_thermodenitrificans_1, _2, _3 and _4, Paenibacillus_barengoltzii_3, Paenibacillus_ginsengihumi_1, Paenibacillus_ihumii_1 and _2, Paenibacillus_lactis_1 and _2, Thermobacillus_composti_1 and _2 and Ureibacillus_terrenus_1 (File S2). In Middle phase, 22 ESVs annotated as species associated with potential cellulose degradation were significantly reduced in relative abundance and only 5 were significantly increased in relative abundance, the Clostridiales, Clostridium_colicanis_1, Gracilibacter_thermotolerans_1 and Ruminiclostridium_thermocellum_1, as well as the Streptosporangiales, Thermomonospora_chromogena_1 and Thermomonospora_curvata_1. Similarly, a further substantial reduction of 33 ESVs significantly reduced from Middle to Aged

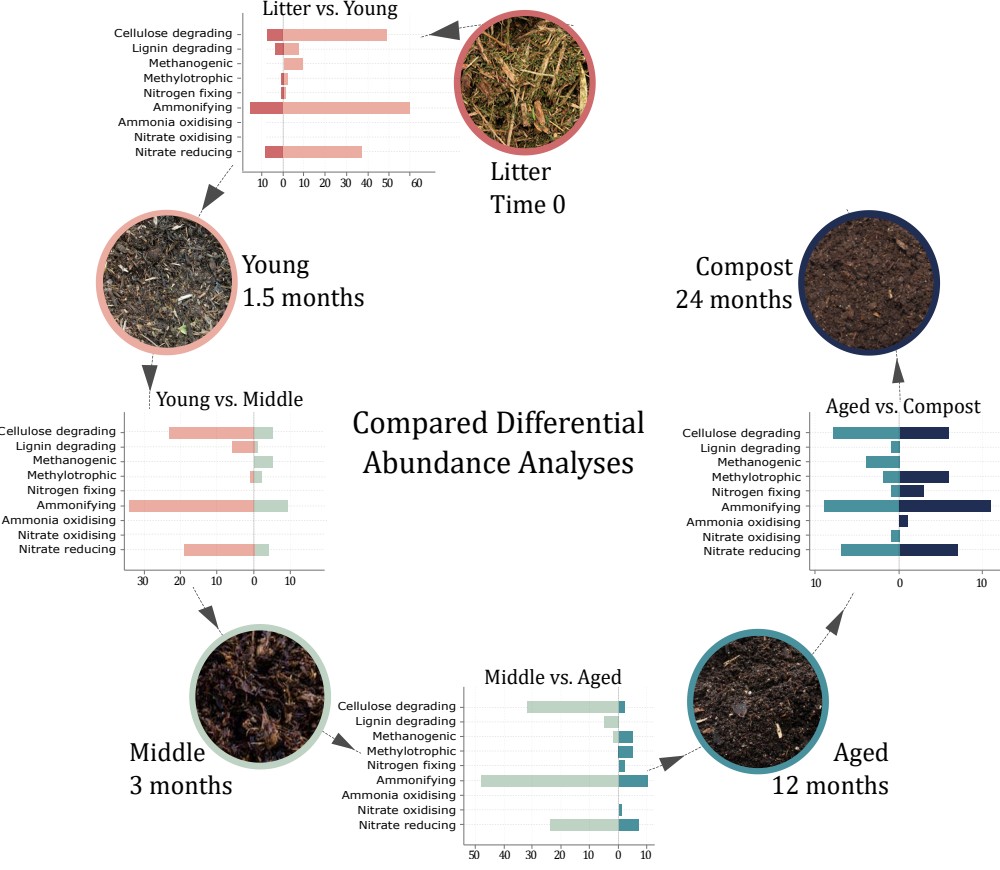

**Figure 5  Number of differentially abundant species involved in the carbon and nitrogen cycle between each successive phase.** Number of differently abundant species between Litter and Young, Young and Middle, Middle and Aged and Aged and Compost associated with cellulose and lignin degradation, methanogenic and methanotrophic activity, nitrogen fixation, ammonification, ammonia and nitrite oxydation and nitrate reduction. Supporting species putative function association, as well as ESV relative abundance, annotation, count distribution, blast statistics, alternative database hits and sequences are provided in File S2. Created using www.visme.co.

phase, while only three increased. Aged phase and mature Compost were relatively similar with only seven ESVs annotated as species associated with putative cellulose degradation significantly increasing: Actinotalea_ferrariae_1, Devosia_honganensis_1, Flavobacterium_degerlachei_1, Hyphomicrobium_zavarzinii_1, Lapillicoccus_jejuensis_1, Nocardioides_aestuarii_1 and Sorangium_cellulosum_2.

Thirteen differentially abundant ESVs were annotated as species associated with lignin degradation potential, belonging to the phyla Actinobacteria (5), Proteobacteria (2) and Firmicutes (6) (Fig. 5). Of these, four ESVs were significantly reduced in relative abundance from Litter to Young phase (Comamonas_testosteroni_1, Gordonia_paraffinivorans_1, Mycobacterium _thermoresistible_1 and Rhodococcus_zopfii_1) while seven ESVs significantly increased in relative abundance (Bacillus_benzoevorans_1, Brevibacillus_borstelensis_1, Brevibacillus _thermoruber_1, Brevibacillus_thermoruber_2,

Ochrobactrum_intermedium_1, Thermomonospora_curvata_1 and Ureibacillus_terrenus_1). Five ESVs were significantly reduced from Young to Middle phase, while only one ESV was significantly increased, (Thermomonospora_curvata_1). Six ESVs annotated as species associated with lignin degradation potential were differentially abundant between Middle and Aged phases, all of which were significantly reduced in relative abundance in Aged phases, Brevibacillus_borstelensis_1, Gordonia_paraffinivorans_1, Thermobifida_cellulosilytica_1, Thermomonospora_curvata_1, Ureibacillus_terrenus_1 and Ureibacillus_thermosphaericus_1, while no significant changes were identified between Aged phase and mature Compost.

### Methanogen and methylotroph community

Thirteen ESVs annotated as putative methanogenic species were differentially abundant between composting phases (Fig. 5, File S2) and belonged to the Methanosarcinaceae (4), Methanosaetaceae (2), Methanomicrobiaceae (3), Methanobacteriaceae (3) and Methanocellaceae (1). Nine of these ESVs were significantly increased in relative abundance from Litter to Young phase, Methanosarcina_MS_1, 2, 3 and 4 (four distinct ESVs which are ambiguous for multiple species within *Methanosarcina*), Methanoculleus_MS_1, Methanoculleus_thermophilus_1, Methanoculleus_hydrogenitrophicus_1, Methanobacterium_formicicum_1 and Methanocella_arvoryzae_1 (File S2). None of these ESVs significantly decreased in relative abundance between Young and Middle phases, but Methanobacterium_MS_2 and Methanothrix_thermoacetophila_1 significantly increased in addition to further significant increases in Methanoculleus_hydrogenitrophicus_1, Methanoculleus_thermophilus_1 and Methanosarcina_MS_3 and 4. The ESVs Methanoculleus_thermophilus_1 and Methanosarcina_MS_4 significantly decreased in relative abundance between Middle and Aged phases, while five ESVs significantly increased, namely Methanobacterium_formicicum_1 Methanobacterium_MS_1 and _2, Methanoculleus_MS_1 and Methanosaeta_concilii_1. There were significant decreases in ESVs annotated as the putative methanogenic species Methanobacterium_formicicum_1 and Methanothrix_thermoacetophila_1, as well as Methanosarcina_MS_2 and 3, from Aged phase to mature Compost, but no significant increases.

The differently abundant ESVs annotated as putative methylotroph species belonged to three families within Proteobacteria: Methylococcaceae (7), Methylocystaceae (4) and Rhodobacteraceae (1). One ESV was significantly reduced progressively in relative abundance between Litter, Young and Middle phase, Paracoccus_kondratievae_1, while two ESVs significantly increased, namely Methylocaldum_szegediense_1 and Methylocaldum_tepidum_1 (Fig. 5, File S2). No ESVs annotated as putative methylotroph species significantly decreased in relative abundance between Middle and Aged phases but five ESVs significantly increased, Methylobacter_MS_1, Methylocaldum_marinum_1, Methylocaldum_tepidum_1, Methylomicrobium_MS_1 and Methylosinus_trichosporium_1. Both Methylocaldum_marinum_1 and Methylocaldum_tepidum_1 subsequently significantly reduced in relative abundance between the Aged phase and mature Compost, while six ESVs annotated as putative methylotroph species significantly increased, Methylobacter_marinus_1, Methylobacter_MS_1 and

Methylosarcina_lacus_1, as well as the Rhizobiales Methylocystis_echinoides_1, Methylocystis_rosea_1 and Methylocystis_MS_1.

## Microbes associated with nitrogen cycling
### Ammonification and Nitrification
ESVs annotated as putative nitrogen cycle-associated species and differentially abundant between composting phases included 108 species associated with ammonification, one with ammonia oxidation, one with nitrate oxidation, 67 with nitrogen reduction, and six with nitrogen fixation (Fig. 5; File S2).

ESVs annotated as species associated with putative ammonification belonged to the phylum Firmicutes (46), Actinobacteria (37), Proteobacteria (19), Bacteroidetes (4), Chloroflexi (one) and Euryarchaeota (one). Sixteen of these ESVs were significantly reduced in relative abundance from Litter to Young phases, while 70 significantly increased. The ESVs that have significantly increased in Young phase were dominated by Bacillaceae (15) and Paenibacillaceae (nine) but the largest fold change increases were observed in the Bacteroidetes Rhodothermus_profundi_1, the archaeon Methanoculleus_thermophilus_1 and the Proteobacteria Legionella_londiniensis_1. Thirty-six ESVs significantly reduced from Young to Middle phases and only nine increased, including the Bacillales Caenibacillus_caldisaponilyticus_1 and Paenibacillus_yonginensis_1, as well as further significant increases in Methanoculleus_thermophilus_1. Significant reductions of 49 ESVs annotated as species potentially associated with ammonification occurred from Middle to Aged phases, while 10 ESVs increased. Aged phase and mature Compost were more similar, within only eight ESVs significantly lower in relative abundance and 11 significantly higher, with the largest fold change increases being within Flavobacterium_degerlachei_1 and Lysobacter_yangpyeongensis_2.

Only one differently abundant ESVs annotated as a species with ammonia oxidation potential was identified, the ammonia-oxidizing archaea Nitrosotenuis_cloacae_1, which significantly increased in relative abundance from Middle to Aged phase and was then at a lower relative abundance in Compost compared to Aged. Some differently abundant methane-oxidizing bacteria could also have the potential to oxidize ammonia. This includes Methylocaldum_szegediense_1 and Methylocaldum_tepidum_1 which were also at a higher relative abundance in Aged compared to Compost while Methylobacter_marinus_1 and Methylocystis_echinoides_1 increased in relative abundance from Aged to Compost. Similarly, only one differently abundant ESVs annotated as a putative nitrite oxidizing species was identified, Nitrospira_japonica_1, which significantly increased in relative abundance in mature Compost phase compared to Aged.

### Denitrification and nitrogen fixation
Sixty-seven differently abundant ESVs could be annotated as species putatively associated to denitrification (File S2). These included species within Firmicutes (31), Actinobacteria (16), Proteobacteria (13), Bacteroidetes (three), Chloroflexi (one), Ignavibacteriae (one), Deinococcus-Thermus (one) and Planctomycetes (one). Nine differently abundant ESVs were annotated as species associated with denitrification potential and were significantly reduced in relative abundance from Litter to Young phase, whereas 37 ESVs significantly

increased, the majority being from Bacillales (22), but the largest fold change increases was observed in Rhodothermus_marinus_1 and _2, Thermogutta_terrifontis_1 and Thermus_thermophilus_1 (Fig. 5). This was followed by a significant decrease in relative abundance for 20 ESVs and an increase in four ESVs from Young to Middle phase, the Firmicutes, Bacillus_smithii_1, Clostridium_colicanis_1, Paenisporosarcina _macmurdoensis_1 and Thermomonospora_chromogena_1. Twenty-four ESVs significantly decreased in relative abundance from Middle to Aged phase, including 13 ESVS from Bacillales, such as Geobacillus_thermodenitrificans_1, _2, _3 and _4, Thermoactinomyces _khenchelensis_1 and Pseudoxanthomonas_taiwanensis_1, while six ESVs increased in the Aged phase compared to Middle, with the largest fold change increase observed in Hyphomicrobium zavarzinii_1 and Ignavibacterium_album_1. Seven ESVs annotated as species associated with potential denitrification decreased in relative abundance from Aged phase to mature Compost and seven significantly increased, the Proteobacteria Comamonas_aquatica_1, Geobacter_thiogenes_1 Hyphomicrobium_zavarzinii_1, Methylosarcina_lacus_1, and Pseudomonas_aeruginosa_1 as well as the Actinobacteria Actinotalea_ferrariae_1 and Firmicutes Paenisporosarcina_macmurdoensis_1.

Six differently abundant ESVs from species associated with putative nitrogen fixation were observed during the composting process. Cellvibrio_diazotrophicus_1 was present at a significantly higher relative abundance in Litter compared to Young, whereas the archaea Methanobacterium_formicicum_1 significantly increased in relative abundance from Litter to Young phase and again from Middle to Aged phase. The methylotroph Methylosinus_trichosporium_1 also significantly increased in relative abundance from Middle to Aged phase and the three methylotrophs Methylobacter_marinus_1, Methylocystis_echinoides_1 and Methylocystis_rosea_1 significantly increased in the mature Compost compared to the Aged phase.

# DISCUSSION

## Physiochemical changes throughout composting

The high recorded temperature during Young (67.8 °C) and Middle (62.1 °C) phases were accompanied by a significant decrease in 54% of the amount of organic matter at the beginning of the composting process (Fig. 2). The substantial changes from Litter to Middle phase suggest that most of the organic matter decomposition occurred within the first three months, under thermophilic conditions. The thermophilic phase is often considered as the most microbially active as the high temperatures increase the reaction rates and the efficiency of thermostable enzymes to reach an optimal level of lignocellulose degradation (Ryckeboer et al., 2003).

The three main plant structural components; cellulose, hemicelluloses and lignin were quantified at each phase to monitor their degradation (Fig. 2). For all three, the sharpest decrease occurred between Litter and Young, with a 57.5% decrease in hemicellulose, 66.3% decrease in cellulose and a 43.0% decrease in lignin. A significant decrease between Young and Middle was observed for cellulose, but not for hemicelluloses and lignin. These results suggest that the majority of lignocellulose degradation occurred during the

thermophilic stages, which is consistent with expected temperatures of 55−60 °C for optimal lignocellulolytic activity in composting (*Tuomela et al., 2000*).

Ammonium increased significantly in Young before a significant decrease was observed in Middle, while $NO_2^-$-$NO_3^-$ content increased between Litter and Middle phase before decreasing significantly between the Middle phase and Compost (Fig. 2). The increase in $NH_4^+$ content corresponds to high reductions in organic matter, here being plant material and horse feces, and is expected during the thermophilic phase (*Bernal, Alburquerque & Moral, 2009*). The following decreases in $NH_4^+$ could be due to uptake by microorganisms, direct oxidation by ammonia oxidizing microorganisms (AOB/AOA), or through loss by volatilization of $NH_3$. The accumulation of $NO_2^-$-$NO_3^-$ in the following phases and high reduction in total nitrogen suggests ammonia oxidation as well as volatilization may be occurring. Nitrate content is a major criterion of compost quality (*Bernal et al., 1998*), so the drop in $NO_2^-$-$NO_3^-$ concentrations during the later Aged phase and mature Compost could be considered as deleterious to high quality compost production.

## Microbial community structure and composition through composting

The most diverse and abundant bacterial phyla; Proteobacteria, Firmicutes, Actinobacteria, Bacteroidetes and Chloroflexi (Fig. 3A), are commonly reported as dominant in compost, ranging from 72% to 92% of microbial diversity (*Antunes et al., 2016*; *Partanen et al., 2010*; *Wei et al., 2018*; *Zhou et al., 2018*). A total of 299 distinct putative species of bacteria and archaea were identified across the different composting phases. Of these, 50 are shown in Fig. 4. These 50 ESVs are found to have the highest relative abundance across all composting phases and belong to 35 different genera, the majority of which are in the phyla Firmicutes (13), Actinobacteria (eight) and Proteobacteria (eight). Moreover, two of the three species identified here as having highest relative abundance across all samples, *Thermobifida fusca* and *Thermobifida bifida*, agree with the dominant genera, *Thermobifida* and *Thermopolyspora,* found by *Zhang et al. (2015a)* and *Zhang et al. (2015b)* within maize straw compost. This high diversity illustrates the complexity of compost systems, although the substantial amount of remaining unknown or poorly characterized ESVs (Fig. 3B) indicates the extensive amount of natural diversity left to be explored in everyday biological systems, such as compost.

Ordination of samples suggested substantial differences in the microbial community between each composting phase (Fig. 3E). Bacterial diversity increased as composting phases progressed, with few species with high relative abundance in early phases and many species present at a lower abundance in the mature compost (Fig. 3D). Previous research has suggested that limited resources within mature compost can create strong competition and limit diversity of bacteria (*Antunes et al., 2016*; *De Gannes, Eudoxie & Hickey, 2013*). However, diversity indices reported here suggest that high temperatures are likely to have created stronger selection and limited diversity during thermophilic phases, as observed elsewhere (*Ryckeboer et al., 2003*; *Zhou et al., 2018*). This was further supported during differential abundance (DA) analysis, which revealed that the largest numbers of species changes occurred between Litter and Young, and Middle and Aged phases (Fig. 3F, File S2), corresponding to the largest shifts in temperature. These general changes can be attributed

to several factors, but temperature, pH, and OM content, including total carbon and nitrogen, are probably the major drivers of composting microbial community changes, as presented in Fig. 4.

Species reduced from Litter to Young were consistent with the rapid increase in temperatures from 38.2 °C to 67.8 °C, with largest reductions in previously reported mesophilic species such as *Sphingobacterium jejuense*, *Sandaracinus amylolyticus*, *Nakamurella panacisegetis* and *Sphingobacterium cibi* (*Kim, Lee & Lee, 2012*; *Lai et al., 2016*; *Mohr et al., 2012*; *Siddiqi et al., 2016*) (File S2). The correlation analysis (Fig. 4) indicates a strong effect of temperature on the relative abundance of the main ESVs during the Young phase, suggesting a selection effect of temperature on the species composition. Reductions in species such as *Delftia litopenaei* also suggests an analogous dynamic in organisms not captured by the 16S rRNA gene amplification, such as insects, as *Delftia* are dominant gut symbionts in arthropods common to composts (*Morales & Wolff, 2010*; *Wang et al., 2014*; *Xie et al., 2012*). Expected major increases in extremely thermophilic bacteria also occurred, including *Thermomicrobium carboxidum*, *Rhodothermus profundi*, *Rhodothermus marinus*, *Thermogutta terrifontis*, *Thermus thermophilus* and *Thermoleophilum album*, which were all originally isolated from geothermally heated biofilm, hydrothermal vents or hot springs (*Alfredson et al., 1988*; *King & King, 2014*; *Marteinsson et al., 2010*; *Oshima & Imahori, 1974*; *Slobodkina et al., 2015*; *Zarflla & Perry, 1984*) and some of which have previously been reported in compost systems at the specie or genus level (*Antunes et al., 2016*; *Gladden et al., 2011*; *Varma, Dhamodharan & Kalamdhad, 2018*).

Subsequent reductions of species from Young to Middle phases included further loss of thermosensitive species such as *Jeotgalicoccus psychrophilus*, as well as animal-associated *Corynebacterium* sp., *Kurthia massiliensis* and *Streptococcus equinus* (*Bernard et al., 2010*; *Roux et al., 2012*; *Schlegel et al., 2003*; *Yoon et al., 2003*), which likely represents reductions in species associated with horse feces present in Litter. The major species increasing in Middle phase included thermophiles common to compost, such as *Caenibacillus caldisaponilyticus* (*Tsujimoto et al., 2016*) and unexpected species such as the thermophilic and alkaliphilic Verrucomicrobia *Limisphaera ngatamarikiensis* (*Carere et al., 2020*), but was characterized by significant increases in thermophilic methanogenic archaea, the largest change being observed in *Methanothrix thermoacetophila* (*Kamagata et al., 1992*). *M. thermoacetophila* produces methane under lower oxygen conditions such as those of progressing composting of Middle phase and which likely led to the subsequent increases in moderately thermophilic methanotrophs such *Methylocaldum szegediense* (*Medvedkova et al., 2009*).

The transition from Middle to Aged phases saw the greatest shift in differentially abundant taxa as well as an increase in pH and substantial decrease in temperature, with a drop from 62.1 °C to 46.1 °C. These physicochemical variations most likely played a role in microbial proliferation, whereas the abundant ESVs in Aged were strongly correlated with pH and had a weak negative correlation with temperature (Fig. 4). This led to subsequent large reductions in the thermophilic species which had increased during Young and Middle phases, including *T. carboxidum*, *R. profundi*, *T. terrifontis* and *L. ngatamarikiensis,* as well as reductions in thermophilic archaea such as *Methanoculleus thermophilus* (File S2 and

Supplementary DA figures). The significant reduction in *L. ngatamarikiensis* suggests that the mesophilic Aged phase is likely highly competitive despite reaching an optimal alkaline condition for *L. ngatamarikiensis* of pH 8.42 (*Carere et al., 2020*), and illustrates how specific species can be highly transient throughout each composting phase as the ESV Limisphaera_ngatamarikiensis_1 was absent (below detection) in Litter and mature Compost samples. Species which substantially increased in the Aged phase included mesophilic archaea, potentially replacing lost thermophilic archaea in similar niches, such as methanogen *Methanosaeta concilii* and the ammonia oxidizing *Nitrosotenuis cloacae* (*Li et al., 2016*; *Patel & Sprott, 1990*), which corresponds to the reduced ammonia and increased nitrites/nitrates in middle and Aged phases. Species substantially increasing also included the metabolically flexible *Ignavibacterium album* (*Iino et al., 2010*) which, as a generalist, could be benefiting from the extreme disturbance associated with the transition from a thermophilic to mesophilic habitat (*Chen et al., 2021*). The cross-feeding or syntrophic archaea *Methylocaldum marinum* and bacteria *Syntrophobacter sulfatireducens* and *Sulfurivermis fontis* (*Chen, Liu & Dong, 2005*; *Plugge et al., 2011*; *Takeuchi et al., 2019*) also substantially increased, illustrating the potential advantage provided to cooperative strategies in highly diverse and competitive environments such as Aged phase (*Hibbing et al., 2009*).

Further reductions in thermophilic archaea and bacteria occurred between Aged phase and mature Compost, including *M. thermoacetophila*, *T. terrifontis*, *R. profundi* and *R. marinus* as well as the transient *Ignavibacterium album* and the extremophile and nematode pathogen *Leucobacter chromiireducens* (*Muir & Tan, 2008*). Within highly diverse mature compost, large increases were observed in specialized species such as the obligate methanotroph *Methylosarcina lacus* and the nitrite-oxidizing bacteria *Nitrospira japonica* (*Fujitani et al., 2020*; *Kalyuzhnaya et al., 2005*), but the community was best characterized by increases in 51 ESVs annotated as taxa within the order Rhizobiales which increased when compared to Aged phase, such as the methanotroph *Methylocystis rosea*, the nodule associated *Mesorhizobium tamadayense* and *Rhizobium helanshanense*, suggestive of a compost community which could benefit soil health and plant rhizosphere associations (*Qin et al., 2012*; *Rahalkar et al., 2018*; *Ramírez-Bahena et al., 2012*).

## Microbes associated with carbon dynamics
### Lignocellulose degradation early in composting

Cellulose is the most abundant plant polysaccharide and represents an important source of carbon for microorganisms within composting. The bacteria associated with the potential for cellulose degradation (cellulase and/or β-glucanase activity) were concentrated in the thermophilic Young and Middle phases (Fig. 5), although overall, the abundance of lignocellulose is not particularly correlated with the relative abundance of the most abundant ESVs in these two phases (Fig. 4). Increasing in the Young phase, this group included the Bacillales species: *Bacillus borbori*, *Bacillus coagulans*, *Brevibacillus thermoruber*, *Brevibacillus borstelens* is, *Cohnella panacarvi*, *Ureibacillus terrenus*, *Thermobacillus composti*, as well as four *Paenibacillus* and three *Geobacillus* species (*Ali, Hemeda & Abdelaliem, 2019*; *Makky, 2009*; *Odeniyi, Onilude & Ayodele, 2009*;

*Ting et al., 2013*; *Togo et al., 2016*; *Wang et al., 2013*; *Watanabe et al., 2007*; *Zainudin et al., 2013*), while increases from Young to Middle phase were limited to the species *Clostridium colicanis*, *Gracilibacter thermotolerans* and *Ruminiclostridium thermocellum* from Clostridiales, and *Thermomonospora curvata* and *Thermomonospora chromogena* from Streptosporangiales (File S2) (*Chertkov et al., 2011*; *Greetham et al., 2003*; *Sheng et al., 2016*; *Wu et al., 2018*). Within the plant cell wall matrix, cellulose is recalcitrant to deconstruction and requires the sequential action of enzymes (endoglucanases, cellobiohydrolases and β-glucosidases) for glucose liberation (*Béguin & Aubert, 1994*), which are characterized by higher efficiency under thermophilic conditions (*Tuomela et al., 2000*). Although a range of biotic and abiotic interactions could underlie these microbial shifts, the cellulose degradation associated species increasing in thermophilic Young and Middle phases were largely those with characterized optimal temperatures of 55−60 °C, such as *T. composti*, *B. thermoruber* (*Yildiz et al., 2015*), *Paenibacillus barengoltzii*, *T. curvata* and *T. chromogena* (*Padden et al., 1999*; *Satyanarayana, Kawarabayasi & Littlechild, 2013*; *Watanabe et al., 2007*; *Zainudin et al., 2013*). Widespread reductions in cellulose degradation associated species occurred from Middle to Aged phase and mature Compost, but still a few species had a higher relative abundance, such as *Actinotalea ferrariae*, *Flavobacterium degerlachei*, *Lapillicoccus jejuensis*, *Nocardioides aestuarii* and *Sorangium cellulosum* as well as the Rhizobiales species *Devosia honganensis* (*Lee & Lee, 2007*; *Li et al., 2013*; *Schneiker et al., 2007*; *Van Trappen et al., 2004*; *Yi & Chun, 2004*; *Zhang et al., 2015b*), some of which have characterized optimal temperatures of <35−40 °C (*A. ferrariae* and *N. aestuarii*).

Species known to secrete one or many lignin-modifying enzymes were observed throughout the composting phases (Fig. 5), such as the dye-decolourizing peroxidase producers *T. curvata*, *Thermobifida cellulosilytica* and *Mycobacterium thermoresistible*, the lignin-peroxidase producer *Ochrobactrum intermedium* (*Azizi-Shotorkhoft et al., 2016*; *Tian et al., 2016*), as well as *B. borstelensis*, *B. thermoruber*, *Comamonas testosteroni* and *Ureibacillus thermosphaericus* which can produce both peroxidases and laccases (File S2) (*Ndahebwa Muhonja et al., 2018*; *Niu et al., 2021*; *Rashid et al., 2017*). Putative lignin degraders were also concentrated in the early composting stages, with those present in Litter, *C. testosteroni*, *Gordonia paraffinivorans* and *M. thermoresistible*, rapidly depleted or lost during thermophilic phases where thermophilic *Bacillus benzoevorans* (*Wang et al., 2019*), *B. borstelensis*, *B. thermoruber*, *O. intermedium*, *T. curvata* and *U. terrenus* increased. A corresponding significant decrease in lignin content was measured between Litter and Young. The significant decline in the abundance of putative lignin degraders in Aged phase suggests a reduction in lignin degradation, which corresponds to the significant decrease in lignin content measured between Litter and Young, and subsequent stabilization of lignin levels after Middle phase. Lignin is generally considered to be a very recalcitrant and largely degraded by white rot fungi during the compost maturation (*Ryckeboer et al., 2003*). While it is not possible to conclude that lignolytic activity has ceased within Aged phase and mature Compost since the fungal community wasn't characterized, these significant dynamics do highlight candidate bacteria which could play a role in lignin degradation at the beginning of the composting.

## Methanogens and Methylotrophs community

Differentially abundant species identified as methanogens were present throughout the composting phases and belonged to the archaeal families Methanosarcinaceae, Methanomicrobiaceae, Methanocellaceae, Methanobacteriaceae and Methanosaetaceae. The Young and Middle phases hosted both prevalent thermophilic aceticlastic methanogens such as *M. thermophilus and M. thermoacetophila* (*Kamagata et al., 1992*; *Tian, Wang & Dong, 2010*) and thermophilic hydrogenotrophic methanogens, such as *Methanocella arvoryzae* (*Sakai et al., 2010*) as well as the hydrogenotrophic methanogen *Methanoculleus hydrogenitrophicus*, where previously reported isolates were considered as mesophilic (*Tian, Wang & Dong, 2010*) (*Fig. 5*, *File S2*). As temperatures dropped in Aged phase, mesophilic methanogens with characterized strains having optimal growth temperatures of 37−45 °C, such as *Methanobacterium formicicum* and *M. concilii* increased in relative abundance (*Bryant & Boone, 1987*; *Patel & Sprott, 1990*). This shift from thermophilic to mesophilic methanogens suggests replacement of species within the methanogen functional niche linked to temperature, but could also be driven by lignocellulose substrate availability, due to characterized syntrophic interactions between methanogen community members, particularly within *Methanosarcina,* and cellulose-degrading bacteria (*Conrad, 2020*; *Lu et al., 2017*), concentrated within the Young phase. This hypothesis is supported by a strong negative correlation between OM, total carbon and nitrogen and plant tissue constituents with ESV abundance in the Aged and Compost phases (*Fig. 4*). The bacteria are thus likely to be more dependent on the compounds resulting from lignocellulose decomposition than on the lignocellulose itself.

ESVs annotated as methylotroph species from Methylococcaceae, Methylocystaceae and Rhodobacteraceae (all within proteobacteria) were differentially abundant through the composting phases. *M. szegediense* and *Methylocaldum tepidum* significantly increased in Young phase, both of which are considered thermophilic (*Cvejic et al., 2000*) and coinciding with increases in methanogens providing substrate (*Fig. 5*). However, the largest increases in methylotroph species were observed as temperature decreased from Middle to Aged phase, including increases of *Methylocaldum marinum* and *Methylosinus trichosporium*, and further increases in *M. tepidum*. As compost cooled and matured, *Methylobacter marinus* and *M. lacus* increased in relative abundance as well as the Rhizobiales species *M. rosea* and *Methylocystis echinoides*. These methanotrophs are thought to be environmentally sensitive, with species from *Methylocaldum* being replaced by species from *Methylosinus*, *Methylobacter*, and *Methylocystis,* as conditions move from thermophilic to mesophilic temperatures (*Halet, Boon & Verstraete, 2006*). Although the dynamics of methanogens (*Thummes, Kämpfer & Jäckel, 2007*) and methanotrophs (*Chen et al., 2014*) have been studied in compost environments, reports of the co-occurrence of methanogenic and methanotrophic species throughout different composting phases is not common. Syntrophic interactions have been characterized as involving consortia of methanogens, such as *M. formicicum*, with sulphur-reducing bacteria, such as *Desulfotomaculum peckii* (*Song et al., 2019*), which significantly increased in the Young phase, and *S. sulfaliredcuens* (*Ahlert et al., 2016*; *Knittel & Boetius, 2009*), which significantly increased in later composting phases. Understanding these dynamics is important for

predicting the production and oxidation of methane in compost, which can substantially influence greenhouse gas emissions from composting.

## Microbes associated with nitrogen dynamics
### Ammonification and Nitrification

Ammonifying bacteria with the ability to lyse proteins, DNA or other forms of organic nitrogen through the action of exogenous proteases, were diverse and present at all stages. This observation is in line with evidence that ammonia is the preferred nitrogen source for most bacteria throughout composting (*Körner & Stegmann, 2002*). The largest proportion of ammonifiers were, however, concentrated in the early thermophilic phase, as 70 species putatively associated to ammonification increased in relative abundance from Litter to the thermophilic Young phase (Fig. 5), including well-characterized thermophiles such *R. profundi* (*Marteinsson et al., 2010*), the archaeon *M. thermophilus* (*Rivard & Smith, 1982*) and *Legionella londiniensis*, which was first isolated from hots springs in Japan and is commonly found in the environment (*Furuhata et al., 2010*). This is consistent with the high level of ammonia recorded ($\pm$ 310 mg kg$^{-1}$), within the Young phase and the strong correlation between NH$_4^+$ content and ESVs relative abundance in Litter and Young (Fig. 4). Following phases saw large reductions in ammonifiers and a few increases in the Middle phase (*C. caldisaponilyticus*) (*Lin, Yan & Yi, 2018*) and in mature Compost (the mesophilic *Lysobacter yangpyeongensis* and the psychrophilic *F. degerlachei*) (*Van Trappen et al., 2004*; *Weon et al., 2006*).

The only ammonia-oxidizing microbe identified across the composting phases was the archaea *N. cloacae*, which increased in abundance in Aged, then decreased significantly in mature Compost (File S2). *N. cloacae* was isolated in 2015 from a wastewater treatment plant in China and has a growth range between 25−33 °C (*Li et al., 2016*). Ammonia oxidation, where NH$_4^+$ is converted to NO$_2^-$ under aerobic conditions, is the first and rate-limiting step of nitrification and therefore could represent an important bottleneck within the mature compost community. Despite the (non-significant) increase in NO$_2^-$-NO$_3^-$ concentration between Litter and the Middle phases, no well-characterized ammonia oxidizing bacteria nor archaea were identified before the Aged phase. As some versatile methylotrophs can also oxidize ammonia (*Hanson & Hanson, 1996*), it is possible that *M. szegediense* and *M. tepidum,* could have driven ammonia oxidation during the thermophilic phase. The nitrite oxidizing bacteria (NOB) *N. japonica* increased in relative abundance from Aged phase to mature Compost phase (being below detection in other phases) (File S2). The prevalence of a well-characterized NOB within mature compost is unsurprising as nitrification normally occurs during compost maturation at mesophilic stages (*Cáceres, Malińska & Marfà, 2018*); however, similar to other reports (*Jiang et al., 2015*; *López-González et al., 2013*), although the increased NO$_2^-$-NO$_3^-$ concentrations during the Middle phase here suggests nitrification occurred during thermophilic phases. In a recent review, *Cáceres, Malińska & Marfà (2018)* highlighted some of the shortfalls in our understanding of nitrification and concluded that future research should explore uncultured *Nitrospira* bacteria in composting. Intriguingly, uncharacterized ESVs placed within the genera *Nitrospira* (Nitrospira_1 and Nitrospira_2) were differentially

abundant and increased in relative abundance from Young to Middle (File S2). The ESV Nitrospira_1 shared 100% sequence identity with an uncultured soil bacterium (GenBank EF667461.1) and was most closely related to *N. japonica* strain NJ11 (98.01%), while the ESV Nitrospira_2 shared 100% sequence identity with an uncultured soil bacterium (GenBank EU012235.1) and shared most sequence similarity to *Nitrospira* sp. KM1 (97.61%), a novel nitrite-oxidizing *Nitrospira* strain isolated from a drinking water treatment plant (*Fujitani et al., 2020*). While caution is necessary when speculating as to the function based on culture independent sequencing, these putative uncultured bacteria could represent new thermotolerant/thermophilic nitrite oxidizing *Nitrospira* species which could be responsible for the increase in nitrite-nitrate concentrations during the Middle phase.

### *Denitrification and nitrogen fixation*

Denitrification is a complex reaction that can be considered complete, leading to the production of $N_2$, or incomplete, resulting in intermediate nitrogen forms, such as $NO_2^-$, NO and $N_2O$. The largest increase in putative denitrifying bacteria was observed between Litter and Young phases, before a large subsequent decrease from Young to Middle and from Middle to Aged phases, indicating preferential denitrification during early thermophilic phases (Fig. 5, File S2). Potential thermophilic denitrifiers able to perform complete denitrification, such as *R. marinus* and *Sphaerobacter thermophilus*, and incomplete denitrification, such as *Pseudoxanthomonas taiwanensis* (*Wang et al., 2010*), were present in higher relative abundance in Middle compared to Aged. Despite the prevalence of nitrate reduction under anoxic conditions (*Gao et al., 2010*), it has been shown that denitrification can occur in the presence of oxygen. For example, although both *R. marinus* and *S. thermophilus* are considered strictly aerobic, they possess NirK genes that allow the reduction of $NO_2$ to NO and an atypical version of the NosZ which allow the reduction of $N_2O$ to $N_2$ (*Sanford et al., 2012*). Subsequently, mesophiles performing complete denitrification such as *Pseudomonas aeruginosa* and *Hyphomicrobium zavarzinii*, as well as incomplete denitrification, such as *I. album*, were present in higher relative abundance in Aged phase and mature Compost (File S2) (*Amaral et al., 1995*; *Arat, Bullerjahn & Laubenbacher, 2015*; *Braga et al., 2021*; *Liu et al., 2012*; *Sanford et al., 2012*; *Wang et al., 2010*). Moreover, it has been demonstrated that *P. aeruginosa* and *I. album* can grow anaerobically with $NO_3^-$ as the only oxygen source (*Sanford et al., 2012*). Identifying the dynamic changes in the denitrifying microbial community can help understanding of the emissions of nitrous oxide from composting which, as a potent greenhouse gas, represent a risk to the environment. Likewise, while complete denitrification and $N_2$ production is not harmful to the environment, nitrogen loss during composting should be avoided so as not to reduce the agricultural quality of compost.

The proliferation of putative methanotrophic organisms such as *M. marinum* and *M. lacus* in the Aged and Compost phases could also have played a role in the decrease of $NO_3^-$ concentrations after Middle phase (Fig. 5). These bacteria can use $NO_3^-$ as a source of nitrogen as well as suppress nitrifiers through competition for oxygen in low oxygen conditions (*Megraw & Knowles, 1987*) as well as providing carbon for various denitrifiers, such as *H. zavarzinii* (*Amaral et al., 1995*). Similarly, although the free-living nitrogen

fixing bacteria *Cellvibrio diazotrophicus* (*Suarez et al., 2014*) was present in Litter but significantly reduced in Young, the potential for nitrogen fixation was present throughout the composting process due to the methanogen *M. formicicum* (*Magingo & Stumm, 1991*), increasing in relative abundance from Middle to Aged phase, and also due to the increases in certain the methanotrophs which contain nitrogen fixing strains such as *M. trichosporium*, *M. echinoides*, *M. rosea* and *M. marinus* (*Auman, Speake & Lidstrom, 2001*; *Dedysh, Ricke & Liesack, 2004*; *Oakley & Murrell, 1988*; *Wartiainen et al., 2006*). The presence of bacteria capable of nitrogen fixation could impact the nitrogen balance of the composting process but their presence in mature compost also has the potential to influence the long-term nitrogen availability when applied to agricultural soils (*Batista & Dixon, 2019*).

## Mature compost as a microbial reservoir

Although the microbial community was very dynamic between the early composting phases, only a small number of differentially abundant ESVs were observed between Aged and Compost (Figs. 3F and 5). Thus, there was relative stability in this very diverse community that was also reflected in their relative closeness in ordination (Fig. 3E), regardless of the greater temporal gap between these phases (12 months). The mature compost community included species which could provide beneficial functions to agriculture, such as nitrogen fixation and lignocellulose degradation (Fig. 6). Agriculture may account for 25% of global methane emissions, which were about 145 Mt $CH_4$ $y^{-1}$ per year in 2017 (*Smith, Reay & Smith, 2021*). Agricultural waste composting has been extensively explored for mitigation of methane emissions associated with organic decomposition (*Lou & Nair, 2009*) due to putative methanotrophic microbes present within composting processes. The use of mature compost rich in methanotrophic bacteria, such as *M. lacus, M. marinus, M. echinoides* and *M. rosea*, which substantially increased within mature compost here, could provide a benefit by extending this potential community function to increase the soil methane sink or reduce soil methane emissions after the application in agriculture (*Singh et al., 2010*). Any such methane reductions associated with agriculture could help to meet the ambitious target set out in the COP26 Global Methane Pledge of a 30% reduction in global methane emissions (compared to 2020 levels). Similarly, in addition to the more general evidence that compost application can help to suppress crop pathogens (*Bonanomi et al., 2007*; *Bonilla et al., 2012*; *Hoitink & Fahy, 1986*; *Termorshuizen et al., 2006*), some species detected in the mature Compost are considered to be plant growth-promoting bacteria. These include *A. ferrariae, Geobacter thiogenes, Geobacillus thermodenitrificans, Geobacillus stearothermophilus, L. chromiireducens, M. tamadayenses, P. aeruginosa, Paenibacillus yonginensis* and *R. helanshanense,* which have been shown to bestow improved crop nutrient acquisition and/or resistance to different abiotic stresses such as drought, salinity, hydrocarbons, heavy metals and herbicides (*Aguiar et al., 2020*; *Marchant & Banat, 2010*; *Morais et al., 2004*; *Nevin et al., 2007*; *Pieterse et al., 2014*; *Qin et al., 2012*; *Rahalkar et al., 2018*; *Ramírez-Bahena et al., 2012*; *Sukweenadhi et al., 2014*). Although species-level resolved profiling of complex microbial communities is challenging, identification and tracking of these potentially beneficial species to crops could inform our understanding of how compost could improve agricultural soils or the environmental

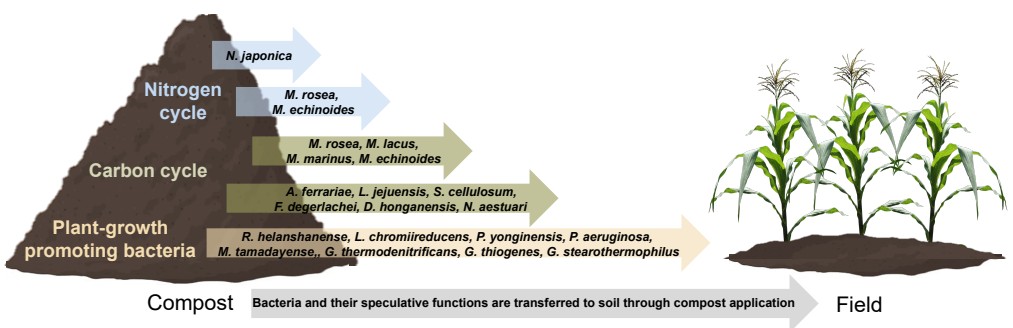

**Figure 6** Representation of the transfer of bacteria and their potential functions from mature compost to soil.

impact of agriculture, in addition to the nutrient and soil stability properties traditionally associated with compost application.

## CONCLUSIONS

As expected, organic matter content gradually reduced over the two years of a windrow-based composting process. Lignocellulose content rapidly reduced while ammonium increased in the first month of composting at the height of thermophilic phase. Nitrite and nitrate concentrations increased later at three months into the process, before also reducing during the cooling phase. Tracking species-level changes in bacteria and archaea across the two-years composting process revealed widespread community shifts through early thermophilic stages, aging mesophilic stages and maturation of compost. Lignocellulose degrading species, including candidate bacteria which could play a role in lignin degradation, were concentrated in the early thermophilic composting stages and corresponded to measured reductions in cellulose, hemicellulose and lignin. Methanogenic archaea and methanotrophic bacteria were present throughout the composting process and were highly dynamic. Similarly, species capable of ammonification and denitrification were present throughout and highly numerous, whereas only a limited number of nitrification bacteria were identified and were concentrated in the final stages of composting. Although the only nitrite-oxidizing species identified, *N. japonica*, was enriched in later maturing compost, a number of poorly characterized sequences sharing close similarity to *Nitrospira* were enriched early thermophilic stages and could represent thermotolerant nitrite-oxidizers. An improved understanding of the dynamics of these methanogenic, methanotrophic and denitrifying populations could help to better control greenhouse gas emissions, such as methane and nitrous oxide, from composting systems and their associated risks to the environment. Similarly, identification of microbial species enriched within the final mature compost included species with well-characterized nitrogen fixing, methanotrophic as well as plant growth promoting activities which could help to inform how compost could provide soils with a rich microbial reservoir of potential benefit to agriculture. A compost heap, although seemingly simple, is a powerful bioreactor that

offers a largely untapped potential for the bioprospecting of under-studied species which could be used in several industries: phytotechnologies, agriculture, agri-food or green chemistry.

## ACKNOWLEDGEMENTS

Special thanks to Jocelyne Ayotte for her help with the physicochemical analyses and Joan Laur for her support. We are grateful to the Montreal Botanical Garden for giving access to their composting platform.

### Funding

This research was made possible by financial support from the NSERC Discovery Grant (FEP RGPIN-2017-05452), Fonds de recherche du Québec –Nature et technologies (FRQNT), Mitacs Acceleration program and Ramo Inc. The funders had no role in study design, data collection and analysis, decision to publish, or preparation of the manuscript.

### Grant Disclosures

The following grant information was disclosed by the authors:
The NSERC Discovery Grant:  FEP RGPIN-2017-05452.
Fonds de recherche du Québec –Nature et technologies (FRQNT).
Mitacs Acceleration program and Ramo Inc.

### Competing Interests

The authors declare there are no competing interests.

### Author Contributions

- Vanessa Grenier conceived and designed the experiments, performed the experiments, analyzed the data, prepared figures and/or tables, authored or reviewed drafts of the article, and approved the final draft.
- Emmanuel Gonzalez analyzed the data, prepared figures and/or tables, authored or reviewed drafts of the article, and approved the final draft.
- Nicholas J.B. Brereton conceived and designed the experiments, authored or reviewed drafts of the article, and approved the final draft.
- Frederic E. Pitre conceived and designed the experiments, authored or reviewed drafts of the article, and approved the final draft.

### Data Availability

   All raw sequence reads are available at NCBI: PRJNA878778.

### Supplemental Information

Supplemental information for this article can be found online at http://dx.doi.org/10.7717/peerj.15239#supplemental-information.

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
