# Peer review of "Dynamics of bacterial and archaeal communities during horse bedding and green waste composting"

_PeerJ, doi:10.7717/peerj.15239_

## Round 0.1 · original submission · Major Revisions

Please provide a point-by-point rebuttal to all the reviewers' comments along with the revised manuscript.

·

Basic reporting

The submitted manuscript presents the results of the analysis of microbiome dynamics during horse bedding composting in five time stages. Analysis of microbiota dynamics was performed using high-throughput sequencing of amplicon libraries of 16S rRNA gene fragment. The interpretation of the study results was made in the context of carbon and nitrogen cycling and greenhouse gas emission.The authors demonstrate a good knowledge of publications on this topic and a well-thought-out approach to setting up the experiment. Unforttunately I am not an expert in English and I can not evaluate the manuscript from this side.
The strength of the work lies in the good experimental material, including the monitoring of a number of physicochemical parameters, including such important ones as сellulose, hemicellulose and lignin content, which is especially important for interpreting the dynamics of the microbiome during composting of a substrate rich in wood raw materials.
The weak side of the work lies in the very formal interpretation of the results, based almost exclusively on the analysis of the differential represented taxa. This part of the work presents a very raw and difficult to read array of data, which is essentially just a listing of differentially represented taxa (of which there are always many in any microbiomes) and information about these taxa found in the current literature. And this is very unfortunate, since I believe that the data obtained in this study is quite enough to make a publication that will be read with great interest.
At the moment results in lines 252-493 and corresponding discussion section are unreadable.
Among other things, I could not find links to raw sequencing data.

Experimental design

The design of the experiment as a whole raises no questions.

Validity of the findings

The main problem of the work is the interpretation of the results. I suggest some ways to improve this part of the work. However, this does not mean that I insist on the use of these approaches. Authors may do as they wish. The manuscript lacks an integral representation and understanding of the relationship between the dynamics of physicochemical parameters and the dynamics of the microbiome.
1) The array of differentially represented taxa contains information about log Fold change values, but not about abundances of taxa. I believe that for this reason a large number of very minor taxa were included in the list.This not only complicates and clutters the discussion, but also introduces taxa into it, whose role can be neglected;
2) I think that the work would benefit from the presentation of a single figure that combines the dynamics of the most important physical and chemical parameters and the dynamics of the most important microbial players;
3) To demonstrate the relationship between parameters and microbial taxa dynamics Canonical correspondence analysis may be useful;
4) To infer function from 16S rRNA sequences not only from literature sources analysis, there are alternative tools like tax4fun and other;

Regardless of which tools and approaches you use, the end result should lead to a more coherent picture that describes the dynamics of the main taxa in relation to the dynamics of parameters, in an accessible form.

Additional comments

137 as well?
219 no information about ambient temperature

Reviewer 2 ·

Basic reporting

The Authors of the manuscript were well descripted their work about the understanding of the complex bacterial community compositions during composting period. A filed experiment was established with three main compost windrows being constructed and four sample timepoints (young, middle, aged and mature compost) were collected to determine the physicochemical properties and the 16s rRNA gene for the bacterial composition.

Experimental design

Really good approach of the experimental design.

Validity of the findings

Only few comments on the statistical approach to improve the visualisation of the results.

Additional comments

As a main a comment for the authors according to my opinion is that they should change at the hole document the phrase “microbial community dynamics or composition” due to the fact that they were investigated only the bacteria communities. Microbial communities are referred to both fungi and bacteria communities, but also for protists and other microorganisms in general. So, I’d like to suggest that if you referring at the manuscript for microbial communities, and especially in your results, needs to be changed to bacterial diversity or bacteria community composition.

A second main comment for the authors is that although the grate job for the determination of the bacterial community composition through the composting period, as it is known also from the authors the main lignocellulose degrading microorganisms are fungi. Bacteria are mainly potential degraders or co-degraders associated with fungi (Chukwuma et al., 2021, doi: 10.3390/ijerph18116001) and I propose that the authors should identify also the fungi diversity from the same samples and correlate the bacteria and fungi communities via a network analysis, using also the physicochemical properties. Just to be clarified the suggestion is for a future work and not being included in this manuscript.

Finally, I’d like to suggest to the authors in future works in order to represent their results from the differential most abundance ESVs in each sample/treatment, to prepare graphs like Heatmaps like those on the following manuscripts:
Bartelme et al., 2019 (http://dx.doi.org/10.1128/mSphere.00143-19)
Nearing et al., 2021 (https://doi.org/10.1101/2021.05.10.443486)

• Could the authors explain why they used the primer set (P609f/P699r) and not the common primer set (515f/806r) which is suggested by the earth microbiome project? The amplicon product is much smaller than the suggested primer set and could help better the taxonomical classification (V3-V4 region of 16s rRNA gene is more hypervariable). With this set of primers, you could totally take advantage the MiSeq run of 2x250 bp paired end.

• Could the authors define the choice of the Mothur pipeline and not the DADA2, which produces the ASVs (amplicon sequencing of variance). ASVs are produced by a unique sequencing that is different by a base or more from the other sequences and could be taxonomicaly classified as a different strain. This approach could solve the problem of using the different databases and the reduction of the threshold of 98% identity/coverage.
• For the PDA analysis (figure 3E) I’d like to propose from the authors to perform a PERMANOVA test in order to identify any statistical differences of the b-diversity differentiation that is observed and the percentage of the observed variances are being explained by the data of the study.
• I would like to suggest to the authors to calculate also other a diversity indexes, such as inverse Simpson (which is associated with the OTUs (ESVs in your case) of intermediate dominance), Fisher’s a (which is related to the highly abundant OTUs) and predicted population richness (ACE for example). It’d would be better understandable if you also indicate with letters the statistical differences on the top of each boxplot in your new figures or in the figure 3D. The new graphs could be added in supplementary figures if there are no statistical differences.
• Could the authors construct a relative abundance stuck barplot of the 100 or 200 most ESVs at phylum level, instead of flower diagram, in order to be clearer the differences in the bacteria composition communities through the five composting phases?
• In figure 2 is not clear visualized the determinations of NH4 and NO2-NO3. Could you please paste them in a different page?
• In the lines 203 - 204 the authors define the statistical analysis of the determinations of the physicochemical properties. Could they please define, if they first run a normal distribution test (e.g. Shapiro test) before to use a parametric test like one way Anova? If the data are not normal distributed, they could perform a Kuskal-Wallis test (no parametric test) to identify the multiple comparisons.
• Line 253: I suggest that the tittle could be rephrase it to Composting Bacteria community compositions.
• Could the authors explain how the identify the differently abundant putative species involved in methane production, lignin degradation, cellulose degradation etc.? Did you use a database for that?
• I suggest to the authors to prepare a correlation heatmap to identify the negative or positive correlations of the physicochemical properties and the 10-20 most abundant ESVs of the dataset in each composting period. It would be interesting if you identify any correlations between those.

Reviewer 3 ·

Basic reporting

In this manuscript, the composition of the components and the bacterial and archaeal communities were analyzed with respect to their gradual changes during composting. The data is very detailed and carefully organized. The results, which focus on the increase or decrease of bacteria and archaea with changes in temperature conditions and composition regarding the five phases, are spectacular, and are fully discussed with many citations. The relationship between bacterial and archaeal communities, and environmental conditions may be useful in assessing the potential functions of compost. The knowledge gained from the detailed data processing will be of much use for future research.
However, although the conclusion focuses on agricultural applications, this manuscript's strengths extend beyond that, as it discusses a variety of perspectives and could focus on a wide range of compost application possibilities. The conclusion that refers only to agricultural use seems to be insubstantial. More possibilities should be discussed, and there must be any features that are unique to horse bedding because the raw material is horse bedding. This manuscript should be revised before publishing in the Journal of Life and Environmental Sciences.
Other points were described as follows, including some minor ones:

1. You should change the title or add a subtitle. Here, horse bedding (wood chips and horse feces) was used, and it is thought that the bacterial communities and its changes will vary depending on what kind of raw material was composted. It would be better to clearly state what kind of material was used in the title.
2. Fig. 2 is cut off and the entire figure cannot be seen.
3. SD lines of the light blue and pink in Fig. 2 are too light in color and difficult to see. Could you make them a little thicker or change the color?
4. The values for NH4+ (µg g-1) (L245) and NO2--NO3- (µg g-1) (L249) are different in Fig. 2 and in the text. Please check them.
5. Please write the formal names of the abbreviations for Taxonomic Level in Fig. 3b and Fig. 3c.
6. L258-259 “331 ESVs unique to one of 299 distinct species.” It is difficult to grasp the meaning of this sentence. Please explain a little more.
7. L270-271 The results of "Young phase compared to Aged phase" are not listed in Supplementary file1.
8. L349 “Flavobacterium _degerlachei_1” Erase spaces. The other names also have space.
9. The figures in Fig. 4 is too small to see. Please make it larger.
10. L559-560 “Reductions in species such as Delftia litopenaei also suggest expected increases in organisms not captured using 16S rRNA gene amplification” &
L606-610 “Further reductions in thermophilic archaea and bacteria occurred between Aged phase and mature Compost hints at alterations between phases in organisms not captured using 16S rRNA gene amplification.”
Why have the authors guessed that the these were not adequately captured by 16S rRNA gene amplification? Please explain in more detail.
11. L647-648 Isn't the sentence here a mistake?
12. L723-727 The sentence here is difficult to read the meaning, please rewrite it.
13. L768-769 Isn't the comma “,” after the citation?
14. L774-775 Please include a citation for “P. aeruginosa and I. album can grow anaerobically with NO3- as the only oxygen source.”
15. L775-779 This sentence is difficult to understand the meaning, please separate the sentences and revise for clarity.
16. Have you submitted your English proofreading? If not, please do so.
17. Conclusion&Figure_5 More possibilities should be mentioned, not only for agricultural use.

Experimental design

Could you write the base length using the forward primer (P609D) and reverse primer (P699R)?

Validity of the findings

The essential composting processes of lignocellulosic degradation, methane and nitrogen cycling are well analyzed, referring to changes in the microbial communities and physicochemical properties. However, only the contribution to agriculture is mentioned, as opposed to the objective of contributing to future research on improving the environment. Please expand the discussion a little more.

Additional comments

Hope the manuscript will be improved and received.

---

## Round 0.2 · Minor Revisions

The paper has been revised appropriately. As I do agree with the comment off the 3rd reviewer in their "additional comments" I ask you to revise accordingly and also make the really minor revisions of all reviewers and clearly address these changes in the revised manuscript and rebuttal.

·

Basic reporting

I have only a few comments on the current version of the manuscript.
69-70 How the understanding of microbial community dynamics could lead to targeted fertilization
with PGP bacteria? The text 861-873 does not make this more understandable;

252-259 I think that these data are quite interesting for the reader, as they describe the dynamics of the
lignocellulosic biomass structure in the composting process. It is possible that the diagram
(see attachment) would be interesting, at least in the supplement;

I think (but do not insist) that in the work there is not a general scheme that summarize the most pronounced features of phases (physico-chemical indicators and taxonomy), which would demonstrate which processes are characteristic of each phase and what changes are accompanied by. It is possible that this picture would better fit instead of the formal graphic abstract that is available now.

Experimental design

OK

Validity of the findings

OK

Additional comments

no

Reviewer 2 ·

Basic reporting

The Authors of the manuscript were well descripted their work about the understanding of the complex bacterial community compositions during composting period. A filed experiment was established with three main compost windrows being constructed and four sample timepoints (young, middle, aged and mature compost) were collected to determine the physicochemical properties and the 16s rRNA gene for the bacterial composition.

Experimental design

As already reported in the previous review, the experimental design of the authors was well organised and described for the aims of the project

Validity of the findings

After the new additional updates of the authors the manuscript is validating the findings.

Additional comments

No additional comments

Reviewer 3 ·

Basic reporting

Thank you for the revisions to the manuscript. The addition of Figure 4 clarified the arguments and the minor text corrections make it very readable.

Experimental design

The sequential analysis focusing on the physicochemical components and the function of microorganisms has potential for development in microbiological research.

Validity of the findings

The following are some of the minor points of concern.

【L40-41】and【L71-73】 
It might be easier to get the meaning of the sentence if you would write "by landfill" here.

【L172-174】 
For the primers, it might be better to state " provide excellent coverage for bacterial and archaeal species" or another. I will leave it to you.

【L310-313】
Would you indicate the table number in which this data was published?

【L844】 VSEs → ESVs mistake?

Additional comments

I would like to suggest that it would be better if you could also add a sentence that the microbial community differs depending on the composting materials, along with some references cited. The intensity of odor at the composting stage varies depending on the composting material, and the changes in the microbial community at the composting stage are also different. Because the process of change in the microbial communities during the composting phase is of great interest, as well as the reason for the change. I leave this to the authors.

---

## Round 0.3 · accepted · Accept

Thank you for this last revision!